# Research on Transformer Omnidirectional Partial Discharge Ultrasound Sensing Method Combining F-P Cavity and FBG

**DOI:** 10.3390/s23249642

**Published:** 2023-12-05

**Authors:** Guochao Qian, Weigen Chen, Kejie Wu, Hong Liu, Jianxin Wang, Zhixian Zhang

**Affiliations:** 1State Key Laboratory of Power Transmission Equipment Technology, School of Electrical Engineering, Chongqing University, Chongqing 400044, Chinaliuhong@stu.cqu.edu.cn (H.L.); wang.jianxin@cqu.edu.cn (J.W.); 2State Grid Tianfu New Area Electric Power Supply Company, Chengdu 610213, China; 3School of Electrical and Electronic Engineering, Chongqing University of Technology, Chongqing 400054, China; zhixian-zhang@outlook.com

**Keywords:** fiber optic sensor, fiber Bragg grating, Fabry-Perot cavity, omnidirectional, partial discharge, ultrasound sensing

## Abstract

To achieve omnidirectional sensitive detection of partial discharge (PD) in transformers and to avoid missing PD signals, a fiber optic omnidirectional sensing method for PD in transformers combined with the fiber Bragg grating (FBG) and Fabry-Perot (F-P) cavity is proposed. The fiber optic omnidirectional sensor for PD as a triangular prism was developed. The hollow structure of the probe was used to insert a single-mode fiber to form an F-P cavity. In addition, the three sides of the probe were used to form a diaphragm-type FBG sensing structure. The ultrasound sensitization diaphragm was designed based on the frequency characteristics of PD in the transformer and the vibration model of the diaphragm in the liquid environment. The fiber optic sensing system for PD was built and the performance test was conducted. The results show that the resonant frequency of the FBG acoustic diaphragm is around 20 kHz and that of the F-P cavity acoustic diaphragm is 94 kHz. The sensitivity of the developed fiber optic sensor is higher than that of the piezoelectric transducer (PZT). The lower limit of PD detection is 68.72 pC for the FBG sensing part and 47.97 pC for the F-P cavity sensing part. The directional testing of the sensor and its testing within a transformer simulation model indicate that the proposed sensor achieves higher detection sensitivity of PD in all directions. The omnidirectional partial discharge ultrasound sensing method proposed in this paper is expected to reduce the missed detection rate of PD.

## 1. Introduction

The transformer is the core equipment of the power transmission system, and the insulation condition is the key to its operation safety. Partial discharge (PD) is an im-portant indicator of insulation deterioration [1]. Carrying out PD sensing helps to de-tect insulation defects in time, thus ensuring the transformer’s safe operation [2,3,4]. Fiber optic sensors have the advantages of high sensitivity, freedom from electromag-netic interference, and can be flexibly installed, so nowadays, their application in transformer condition parameter sensing has received wide attention [5]. The Fabry Perot (F-P) cavity and fiber Bragg grating (FBG) are the two most applied fiber optic sensing techniques for PD ultrasonic sensing [6,7,8,9].

The structure and material of the acoustic-sensitive diaphragm of the fiber-optic ultrasonic sensor determines its sensitivity [10]. In recent years, a large amount of re-search has focused on the development of highly sensitive acoustic diaphragms [11], which has significantly improved their sensitivity. However, there is less optimization for the directionality of the fiber-optic PD ultrasonic sensors. Generally, fiber-optic ul-trasonic sensors can only sense PD ultrasonic signals through the effective vibrating surface of their diaphragm, and their ability to sense PD ultrasonic signals coming to the other sides is weak [12,13]. In the field, PD can occur anywhere inside the trans-former with insulation defects. To avoid missing PD signals as much as possible, achieving high-sensitivity PD detection in all directions is essential.

In recent years, fiber-optic vector acoustic sensors have received widespread attention. These sensors, by spatially combining multiple fiber-optic acoustic-sensitive structures, have expanded the detection range of the sensor, allowing it to collect more acoustic information [14]. For instance, Chen and others wound fibers around compli-ant cylinders made of silicon rubber, forming a tri-component hydrophone [15]. Cranch and others adhered fiber Bragg gratings to cantilever beam structures to create vector acoustic sensors [16]. Liu and others reported a miniature fiber-optic water vec-tor flow sensor based on an array of silicon F-P interferometers [17]. However, apply-ing such research to the ultrasonic detection of PD is still relatively rare. Gao and oth-ers formed a planar lateral array using four fiber-optic F-P probes, but their sensing surface still focused on a single plane, mainly used for PD localization [18].

In response to the above problems, this paper proposes a new fiber optic ultrasonic sensor that can achieve omnidirectional PD ultrasonic detection by combining one F-P cavity and the three FBGs. The acoustic diaphragm was designed based on the fre-quency characteristics of PD in the transformer and the vibration model of the dia-phragm in the liquid environment. Then, the fiber optic omnidirectional sensor for PD with the F-P cavity sensing part and the FBG sensing part was developed, and the fiber optic sensing system for PD was built. Finally, the frequency response characteristics, distance attenuation characteristics, anti-interference performance, directional re-sponse characteristics, and PD detection characteristics of the sensor were tested. The developed sensor in this paper has strong anti-interference performance, and high sen-sitivity and can detect PD signals in all directions, which has good application pro-spects.

## 2. Working Principle

### 2.1. Working Principle of the FBG Sensor

An FBG is equivalent to a narrowband reflector inside the fiber core. The Bragg equation of an FBG can be expressed as [19,20]:(1)$$\lambda_{B}=2n_{eff}$$
where *λ_B_* is the Bragg wavelength, *n_eff_* is the effective refractive index of the fiber core, and *Λ* is the raster period. When the FBG is affected by the ultrasound, its Bragg wavelength will be shifted. The shift of the FBG Bragg wavelength caused by strain and temperature can be described by Equation (2):(2)$$\Delta \lambda =2\left(\Lambda \frac{\partial n_{eff}}{\partial L_{g}}+n_{eff}\frac{\partial \Lambda }{\partial L_{g}}\right)\Delta L_{g}+2\left(\Lambda \frac{\partial n_{eff}}{\partial T}+n_{eff}\frac{\partial \Lambda }{\partial T}\right)\Delta T$$

So the fiber Bragg grating’s Bragg wavelength will shift due to the strain caused by the PD ultrasound, as shown in Figure 1.

### 2.2. Working Principle of the F-P Sensor

The fiber optic F-P sensor is designed according to the multi-beam interference principle. The interference cavity of the extrinsic F-P sensor is formed by the fiber end face and the sensing diaphragm. The F-P interference principle is shown in Figure 2.

After the incident light from the light source reaches the fiber end face, part of the light is reflected to the fiber, and the other part of the light is refracted into the F-P cavity and then refracted and reflected several times on the fiber end face and diaphragm surface, finally returns to the fiber to form a multi-beam interference. The intensity *I_R_* of the interference light can be expressed as:(3)$$I_{R}=I_{0}\frac{R_{1}+R_{2}-2\sqrt{R_{1}R_{2}}\cos\Delta \delta }{1+R_{1}R_{2}-2\sqrt{R_{1}R_{2}}\cos\Delta \delta }$$
(4)$$\Delta \delta =\frac{4\pi nL}{\lambda }$$
where *I*_0_ is the intensity of the incident light, *R*_1_ and *R*_2_ are the reflectivity of fiber end face and diaphragm respectively, Δ*δ* is the phase difference between adjacent beams, *n* is the refractive index of F-P cavity, *λ* is the incident light wavelength, and *L* is the F-P cavity length. From Equations (3) and (4), it can be seen that *I_R_* is mainly determined by the cavity length *L*. The ultrasound generated by PD will change the cavity length *L* and lead to the variation of the interference light intensity.

## 3. Development of the Sensor and Construction of the Sensing System

### 3.1. Development of the Sensor

The structure of the fiber optic omnidirectional sensor probe for PD designed in this paper is shown in Figure 3.

The probe is a triangular prism, with the base being an equilateral triangle, the side length is 20 mm, and the height is 10 mm, which was made of resin material by 3D printing. The hollow structure of the probe is used to insert a single-mode fiber to form the F-P cavity, and the three sides are used to form the FBG sensing part. In this study, a diaphragm-type sensitization encapsulation of the FBG is performed. The FBG is first fixed to the surface of the diaphragm along the radial direction with UV-curable adhesive, and then the diaphragm is fixed to the probe.

To make the sensor with high detection sensitivity of PD, the resonant frequency of the acoustic diaphragm should be reasonably designed according to the frequency characteristics of the PD ultrasound signal. The frequency of ultrasound signal excited by PD in transformer oil is mainly concentrated in 20~200 kHz. There are obvious differences between the ultrasound signal spectrums of different types of PD and a single acoustic diaphragm with a unique resonant frequency makes it difficult to detect PDs of all types [21]. Therefore, this study proposes to design the resonant frequency of the FBG acoustic diaphragm to 20–60 kHz and the resonant frequency of the FP cavity acoustic diaphragm to 60–120 kHz. Sensitive detection of PD is achieved by the ultrasound signal responses of acoustic diaphragms with different resonant frequencies. In this study, the diaphragm works in transformer oil. The resonant frequency of a circular diaphragm in a liquid-phase environment with a fixed perimeter and one side in contact with the liquid is [21]:(5)$$f_{1}=\frac{10.33h}{4\pi a^{2\sqrt{1+0.669\left(\frac{\rho_{m}}{\rho }\right)\left(\frac{a}{h}\right)}}}\sqrt{\frac{E}{3\rho \left(1-\mu^{2}\right)}}$$
where *E*, *ρ*, *μ*, *a* and *h* are Young’s modulus, density, Poisson’s ratio, effective vibration radius and thickness of the circular diaphragm respectively, *m* is the density of the liquid in which the diaphragm is located, in this paper specifically refers to 10# transformer insulating oil (*m* = 895 kg/m^3^). In this study, Corning glass (*E* = 73.6 GPa, *ρ* = 2380 kg/m^3^, *μ* = 0.23) coated with high reflectivity dielectric film was selected as the F-P cavity acoustic diaphragm with dimensions: *a*_1_ = 1.7 mm, *h*_1_ = 0.165 mm, and its theoretical resonant frequency is 82 kHz. Monocrystalline silicon (*E* = 180 GPa, *ρ* = 2330 kg/m^3^, *μ* = 0.278) was selected as the FBG acoustic diaphragm with dimensions: *a*_2_ = 2.5 mm, *h*_2_ = 0.1 mm, and its theoretical resonant frequency is 25.6 kHz. The developed fiber optic omnidirectional sensor is shown in Figure 4.

### 3.2. Construction of Sensing System

The fiber optic sensing system for PD built in this paper is shown in Figure 5.

The system includes the F-P cavity sensing part and the FBG sensing part. The ultra-narrow linewidth laser UNFSR-1550-10-SM-FA-M4 (Discovery Optica, Beijing, China) is used as the light source of the FBG sensing part. The output light of the laser is divided into 3 beams by a 1 × 3 coupler and enters the 3 FBGs, respectively. The returned light reflected by the FBGs is coupled by a 1 × 3 coupler and then inputs to the photodetector. DFB tunable laser AP3350A (APEX Technologies, Marcoussis, France) is used as the light source of the F-P cavity sensing part, and the interference light returned by the F-P cavity inputs to another photodetector. In the sensing system, the splitting ratio of the 1 × 3 optical fiber coupler is 1:1:1, and the photodetector’s model is 2053-F-M. When detecting PD, the wavelength of laser AP3350A is adjusted at the maximum slope of the interference spectrum of the F-P cavity. Since the laser UNFSRL-1550-10-SM-FA-M4 outputs an ultra-narrow linewidth (1 kHz) laser of 1550 nm, the FBGs with a central wavelength of 1550.1 nm and a 3 dB bandwidth of 0.3 nm are used. The output light of the laser is located on the side band of the FBGs reflection spectrum, so the FBG sensing part can work normally. The magnification of the photodetector of the F-P cavity sensing part is set to 100 and that of the FBG sensing part is set to 300.

## 4. Performance Tests of Fiber Optic Omnidirectional Sensor

### 4.1. Frequency Response Test

The dimension of the acoustic diaphragm is designed by Equation (4), and the actual resonant frequency of the diaphragm in transformer oil is determined in this paper by pencil–break experiment. The diaphragm receives the shock waves generated by the broken pencil to do attenuated vibration, and the resonant frequency can be obtained by spectral analysis of the diaphragm vibration response. A steel plate was placed in the transformer oil. The sensor was placed on the steel plate, and an HB pencil with a diameter of 0.5 mm was broken 10 cm from the sensor on the steel plate. The pencil–break impact responses of the F-P cavity acoustic diaphragm and the FBG acoustic diaphragm are shown in Figure 6. There is some difference between the actual value of the resonant frequency of the diaphragm and its theoretical value. The diaphragm is fixed on the probe by UV-curable adhesive and the glue changes the vibration characteristics of the diaphragm, so the resonant frequency of the diaphragm changes.

### 4.2. Directional Response Test

The directional response of the sensor is tested. A spatial spherical coordinate system is established with the sensor as the coordinate origin and the normal of the F-P cavity acoustic diaphragm as the Z-axis. In the spherical coordinate system, take *R* = 15 cm, *θ* = 0°, 15°, 30° … 180° as the circumference and take *φ* = 0°, 30°, 60° … 330° on the circumference in turn to set the source point, as shown in Figure 7. The ultrasonic wave is generated by the signal generator driving the piezoelectric crystal. The signal generator outputs a 2 Vpp sine signal, and the frequency is set to the resonant frequency of the F-P cavity acoustic diaphragm and the FBG acoustic diaphragm in turn. The F-P cavity sensing part and the FBG sensing part are used to detect the ultrasound signal from each sound source point. 20 sets of data are saved at each location for peak-to-peak averaging, and the detection results are shown in Figure 8.

It can be seen from Figure 8 that the F-P cavity sensing part and the FBG sensing part have less fluctuation in the detection results of sound sources at different positions on the same circumference. The sensor has equal amplitude omnidirectionality. The ultrasound signal responses of the sensor to the sound source at different *θ* are different. When the diaphragm is facing the sound source, its amplitude is the largest and sensitivity is the highest. When the diaphragm is placed at a certain angle concerning the sound source, the ultrasound signal cannot be completely received by the diaphragm, and its sensitivity will decrease. 20 sets of data from the F-P cavity sensing part and the FBG sensing part are taken at each *θ* for peak-to-peak averaging. For the F-P cavity sensing part, the detection values of different *θ* are compared to that of 0°. For the FBG sensing part, the detection values of different *θ* are compared to that of 90°. The response decay curves of the two sensing parts in 0°–180° are obtained, as shown in Figure 9.

As can be seen from Figure 9, the F-P acoustic diaphragm and the FBG acoustic diaphragm have a limited range of highly sensitive detection. For F-P acoustic diaphragm, the ultrasonic response has been attenuated to less than 50% when *θ* > 60°; for FBG acoustic diaphragm, the ultrasonic response has been attenuated to less than 40% when *θ* < 30° or *θ* > 150°. A single acoustic diaphragm has an obvious detection blind area and cannot achieve sensitive detection of ultrasound signals in all directions. The fiber optic sensor designed in this paper encapsulates the F-P acoustic diaphragm on the front of the probe and the FBG acoustic diaphragms on the four sides of the probe. The sensor has a total of four acoustic diaphragms for receiving ultrasound signals, widening its highly sensitive detection range, and avoiding the obvious detection blind area.

To verify that the developed fiber optic sensor can achieve omnidirectional sensitive detection of PD in a transformer, the omnidirectional ultrasound signal detection platform was built based on the typical structure of transformers, as shown in Figure 10. Three hollow iron cylinders and two iron plates were placed in the oil tank to simulate the windings and iron yokes respectively. Due to the influence of the solid structure on the propagation of the ultrasound signal, the oil tank was divided into different areas. As shown in Figure 11a, the oil tank was divided into parts A, and B along the midline of its width. As shown in Figure 11b, the oil tank was divided into parts 1, and 2 along the midline of its height. The oil tank was divided into four areas A1, A2, B1, and B2. The omnidirectional fiber optic sensor was placed at the half-height of area A. The piezoelectric crystal was placed in four areas A1, A2, B1, and B2, and the signal generator output pulse signal with center frequency at 70 kHz to simulate PD. The responses of the sensor are recorded, as shown in Figure 11. 

Area 1 is the highly sensitive detection range of the FBG sensing part, and area 2 is the highly sensitive detection range of the F-P cavity sensing part. The ultrasonic waves in area A propagate directly to the fiber optic sensor without blocking, at which time both the FBG sensing part and the F-P cavity sensing part can receive the ultrasound signal. When the piezoelectric crystal was in area A1, the response amplitude of the FBG sensing part was larger and that of the F-P cavity sensing part was smaller. When the piezoelectric crystal was in area A2, the response amplitude of the FBG sensing part was smaller and that of the F-P cavity sensing part was larger. The ultrasonic waves in area B are blocked by the iron cylinder and iron plate during propagation, resulting in energy attenuation. When the piezoelectric crystal was in area B1, the FBG sensing part could detect the ultrasound signal, but its response amplitude decreased significantly, while the F-P cavity sensing part could not detect the ultrasound signal. When the piezoelectric crystal was in area B2, the F-P cavity sensing part could detect the ultrasound signal but the FBG sensing part could not detect it. PD may be generated in any area inside the transformer, and the ultrasonic waves will be attenuated during propagation due to the blocking effect of various solid structures. It is difficult for ordinary F-P sensors and FBG sensors to sensitively detect PD from various locations. The omnidirectional fiber optic sensor for PD developed in this paper combines the ultrasonic responses of the F-P cavity sensing part and that of the FBG sensing part to achieve sensitive detection of PD in all directions. The highly sensitive detection range of the F-P cavity sensing part is complementary to that of the FBG sensing part.

### 4.3. PD Detection Performance Test

Based on the built PD test system, the PD detection performance of the fiber optic sensor is tested. The test system is shown in Figure 12. The size of the oil tank is 0.5 × 0.3 × 0.3 m^3^. The fiber optic sensor was placed inside the oil tank and the PZT was fixed to the outer wall of the oil tank for comparative detection with an amplification gain of 20 dB. The fiber optic sensor and the PZT were placed on different sides of the discharge model, both facing the discharge model and 15 cm away from it.

The lower limit of PD detection of the fiber optic sensor was tested by the metal tip discharge model. The amount of PD was first calibrated. The pulse generator for calibration outputs pulse waves of 50 pC and 100 pC to the discharge model in turn and the output voltage U of the detection device is 60 mV and 120 mV, respectively. By the principle of the pulse current method, the relationship between the PD amount *q* and U is *q* = 0.83 U (unit: mV). After calibration, the lower limit of PD detection is tested for the F-P cavity sensing part and the FBG sensing part respectively. The voltage was increased at a rate of 0.5 kV/s until the fiber optic sensor detected the ultrasound signal, and the detection results of the detection device, fiber optic sensor, and PZT were recorded, as shown in Figure 13.

As shown in Figure 13a, the FBG sensing part detected the ultrasound signal with *U* = 82.8 mV, indicating that its lower limit of PD detection is 68.72 pC, and the response amplitude of the FBG sensing part was significantly higher than that of the PZT. As shown in Figure 13b, the F-P cavity sensing part detected the ultrasound signal with *U* = 57.8 mV, indicating that its lower limit of PD detection is 47.97 pC, and the PZT did not detect the PD signal. The PD detection sensitivity of the fiber optic sensor designed in this paper is higher than that of the PZT.

### 4.4. Distance Attenuation Response Test

The distance attenuation response of the fiber optic omnidirectional sensor was tested in transformer oil. The structure of the test system is shown in Figure 14. The size of the oil tank is 1.0 × 0.5 × 0.5 m^3^. Ultrasonic waves were generated by a signal generator driving a piezoelectric crystal. The signal generator outputs a 2 Vpp sine wave signal with a frequency of 90 kHz. The fiber optic sensor is placed against the inner wall of the oil tank, and the PZT is fixed to the outer wall of the oil tank with an amplification gain of 20 dB for comparison detection. Set the distance between the piezoelectric crystal and the fiber optic sensor (the same for the PZT) as 10, 20, 30…90 cm in sequence. The ultrasound signal responses of the F-P cavity sensing part, the FBG sensing part, and the PZT for each position were recorded separately. 20 sets of data were saved at each position for peak-to-peak averaging, and the detection results are shown in Figure 15. The ultrasound signal responses of the fiber optic sensor and the PZT are approximately exponentially attenuated as the distance increases, and the amplitude of the ultrasound signal detected by the PZT is lower than that of the fiber optic sensor. The reason is that the PZT is fixed to the outer wall of the oil tank, and the ultrasonic waves were received by it through the attenuation effect of the tank wall, resulting in a significant decrease in its amplitude. The internal structure of transformers is complex, and the ultrasound signal generated by PD is blocked by various solid structures (e.g., cardboard, winding) during propagation, causing significant signal attenuation. Therefore, the sensitivity of the fiber optic sensor that can be built into the transformer could be better than that of the external PZT.

### 4.5. Anti-Interference Performance Test

The fiber optic omnidirectional sensor for PD installed inside the transformer is susceptible to the impact of the flowing oil during work. In this paper, the ultrasound signal response of the fiber optic sensor was tested in the case of oil flowing to evaluate the anti-interference performance of the fiber optic sensor, and the test system is shown in Figure 16. Immerse the inlet and outlet of the oil pump into the oil tank and turn on the oil pump to simulate the flowing oil, when the rate is 6.5 L/min. The signal generator outputs a 4 Vpp pulse signal with a frequency of 90 kHz. The ultrasound signal responses of the FBG sensing part and the F-P cavity sensing part were tested in the case of oil pump off and on respectively, and the results are shown in Figure 17. The response waveforms of the fiber optic sensor to the same ultrasound pulse signal vary less with and without the impact of flowing oil. It shows that the fiber optic sensor designed in this paper can achieve effective measurement of ultrasound signal in case of oil flowing, which is highly resistant to interference.

## 5. Conclusions

In this paper, a fiber optic omnidirectional sensing method for PD detection in transformers combined with the FBG and F-P cavity is proposed. The fiber optic omnidirectional sensor for PD was developed and its performance was tested. The following conclusions are obtained:(1)The fiber optic omnidirectional sensor for PD developed in this paper has the F-P cavity sensing part and the FBG sensing part. The resonant frequency of FBG acoustic diaphragm is around 20 kHz and that of F-P cavity acoustic diaphragm is 94 kHz. The sensitivity of the fiber optic sensor is higher than that of the PZT. The lower limit of PD detection is 68.72 pC for the FBG sensing part and 47.97 pC for the F-P cavity sensing part.(2)The directional testing of the sensor and its testing within a transformer simulation model indicate that the sensor’s design, encapsulating four acoustic diaphragms, significantly widened the highly sensitive detection range, ensuring a more comprehensive capture of PD ultrasound signals from all directions.(3)As the distance between the PD source and the sensor increases, the peak-to-peak value of the PD signal detected by the sensor decreases. The rate of decrease is higher in the F-P cavity compared to the FBG. Through the design of different resonant frequencies for the F-P cavity and FBG diaphragm, the sensing needs of high-frequency PD ultrasonic signals and long-distance PD ultrasonic signals are both accommodated. Additionally, the sensor can detect obvious pulsed ultrasound signals in case of oil flowing, which means it has a strong anti-interference ability.(4)In the future, the introduction of more diaphragms with different resonant frequencies can be considered to improve the sensor’s ability to cover the typical acoustic frequencies of PDs. In addition, by combining artificial intelligence algorithms and array signal processing algorithms, this sensor is expected to be used in localization studies of PDs.

## Figures and Tables

**Figure 1 sensors-23-09642-f001:**
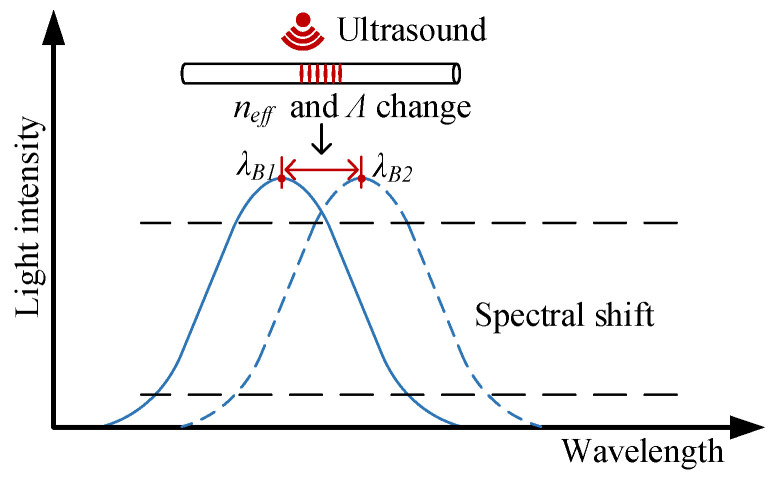
The principle of fiber Bragg grating sensing.

**Figure 2 sensors-23-09642-f002:**
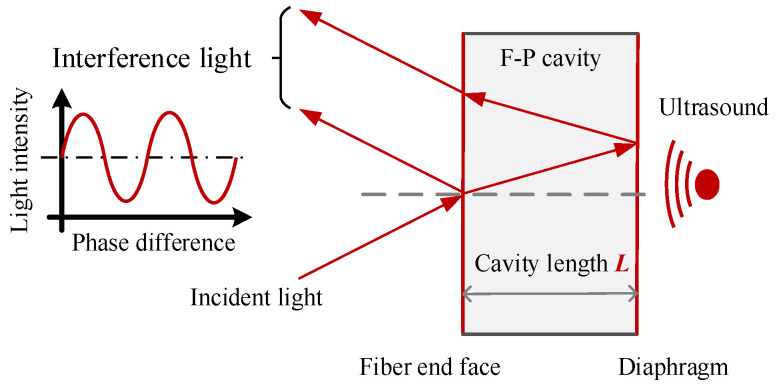
Principle of F-P interference.

**Figure 3 sensors-23-09642-f003:**
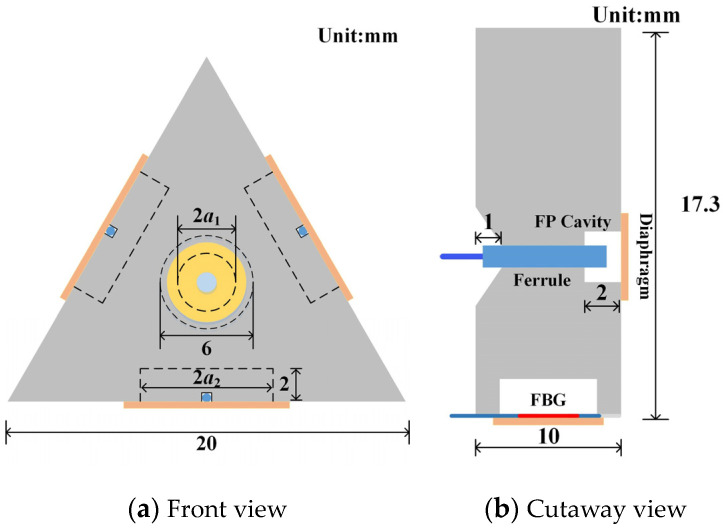
Structure of the fiber optic omnidirectional PD sensor probe.

**Figure 4 sensors-23-09642-f004:**
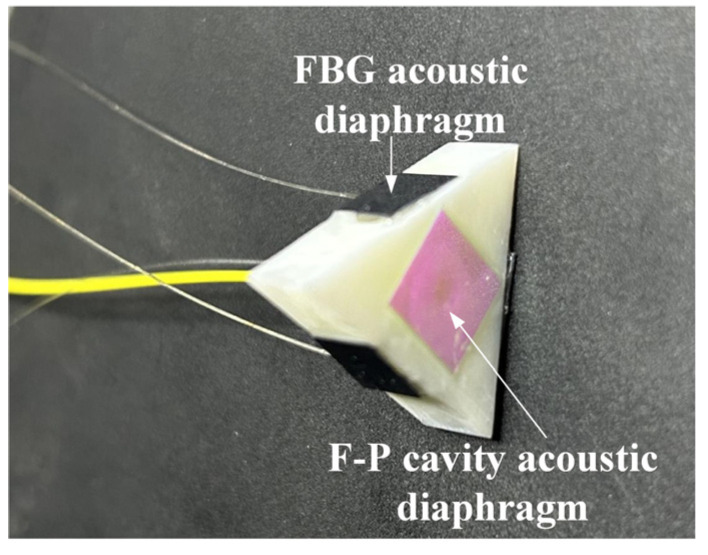
Partial discharge fiber optic omnidirectional sensor.

**Figure 5 sensors-23-09642-f005:**
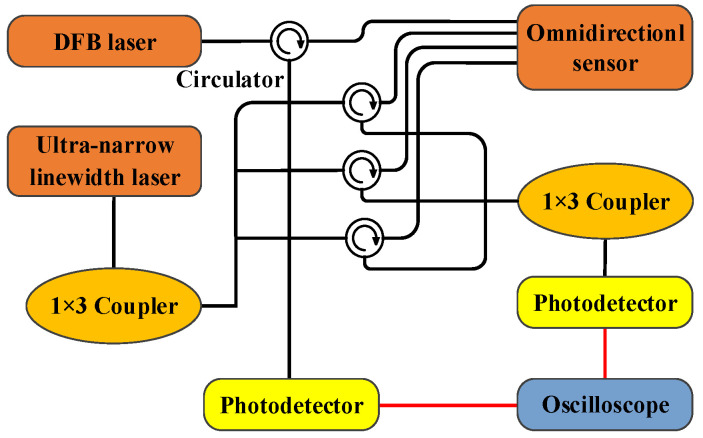
Schematic diagram of the fiber optic sensing system for PD.

**Figure 6 sensors-23-09642-f006:**
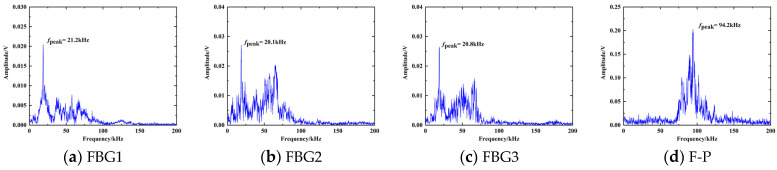
Pencil–break experiment impact response of diaphragms.

**Figure 7 sensors-23-09642-f007:**
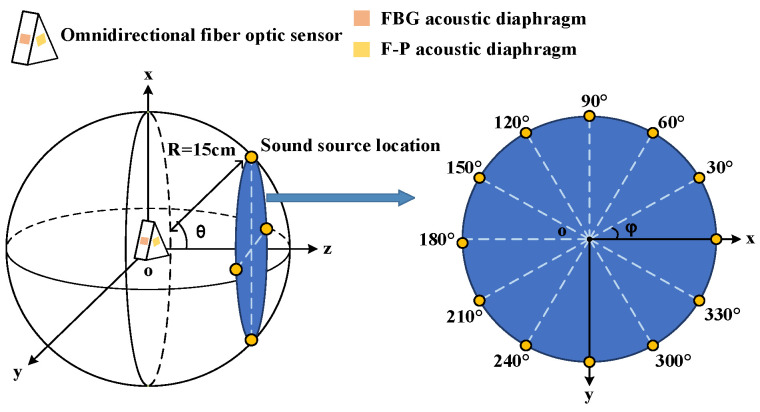
Schematic diagram of directional response experiment.

**Figure 8 sensors-23-09642-f008:**
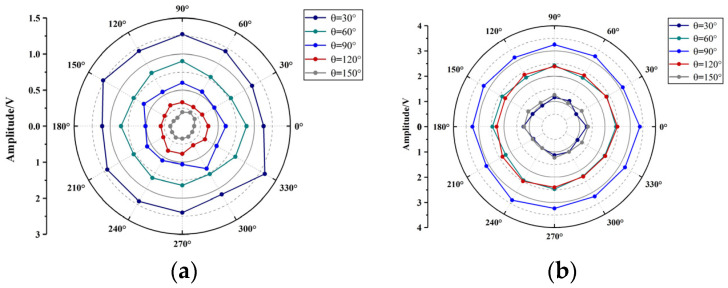
(**a**) Directional response of F-P cavity sensing part; (**b**) Directional response of FBG sensing part.

**Figure 9 sensors-23-09642-f009:**
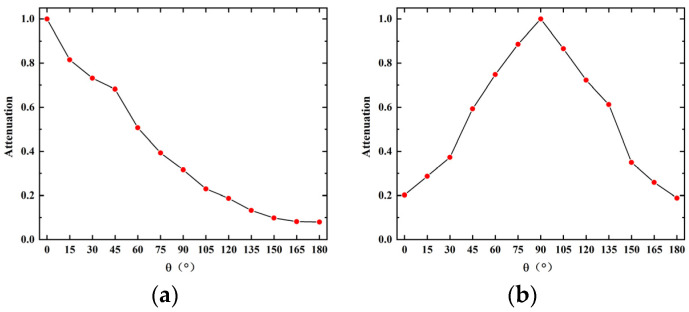
(**a**) Angular decay curve of FP cavity sensing part; (**b**) Angular decay curve of FBG sensing part.

**Figure 10 sensors-23-09642-f010:**
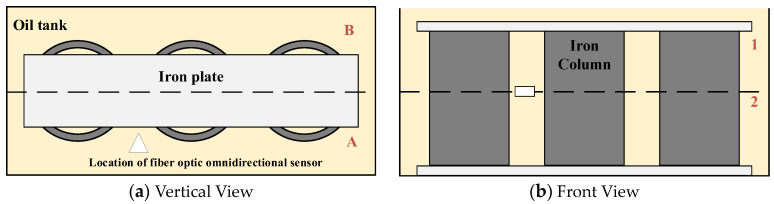
Structure diagram of the omnidirectional ultrasound signal detection platform.

**Figure 11 sensors-23-09642-f011:**
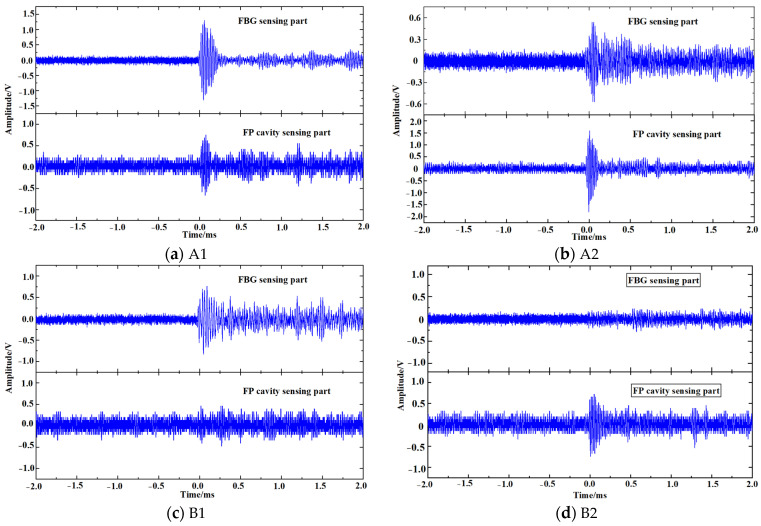
Detection results of the fiber optic sensor for ultrasound signals in different areas.

**Figure 12 sensors-23-09642-f012:**
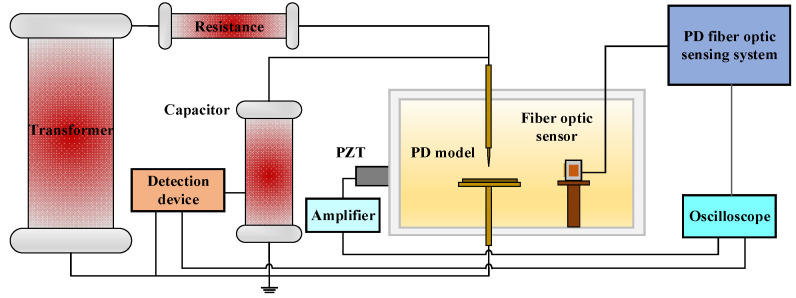
Schematic diagram of PD test system.

**Figure 13 sensors-23-09642-f013:**
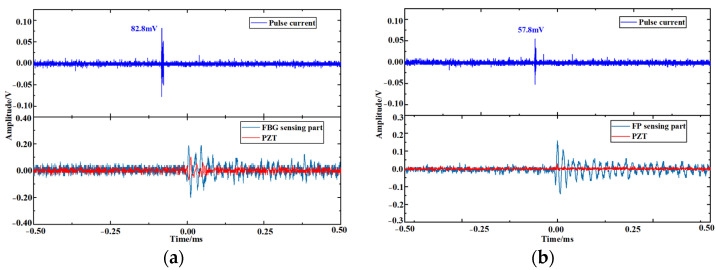
(**a**) The lower limit of PD detection of FBG sensing part; (**b**) The lower limit of PD detection of F-P cavity sensing part.

**Figure 14 sensors-23-09642-f014:**
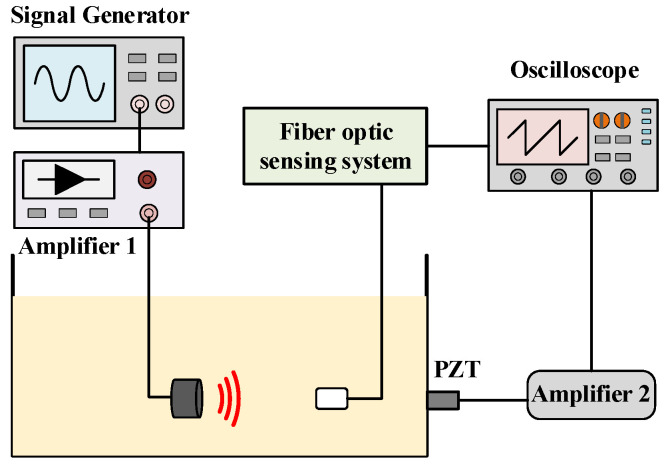
Structure of distance attenuation test system.

**Figure 15 sensors-23-09642-f015:**
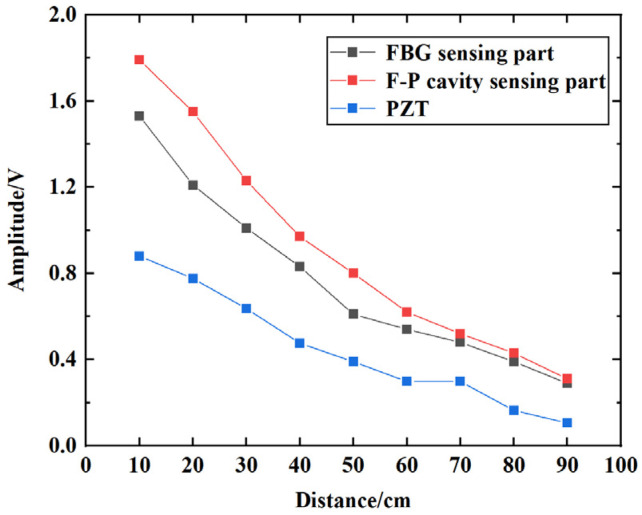
Distance attenuation curve of the fiber optic sensor and the PZT.

**Figure 16 sensors-23-09642-f016:**
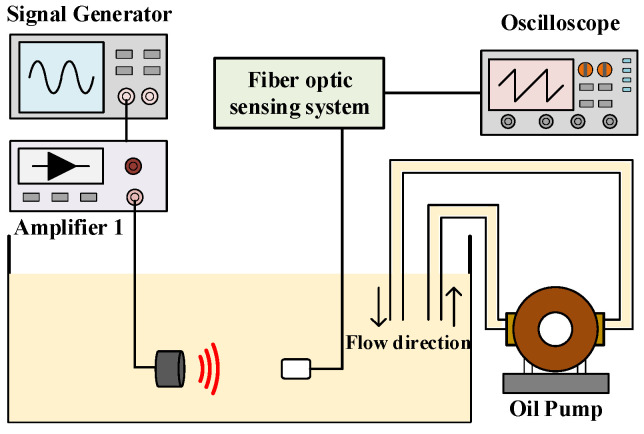
Structure diagram of anti-interference performance test system.

**Figure 17 sensors-23-09642-f017:**
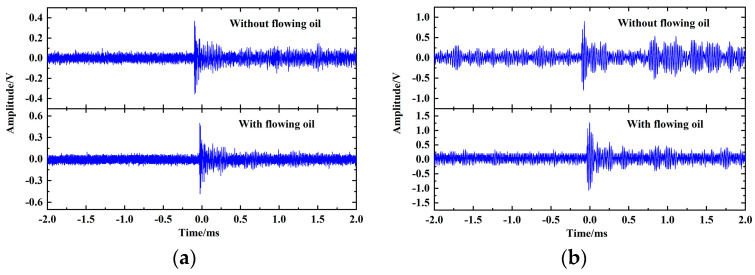
(**a**) The test results of the FBG sensing part; (**b**) The test results of the F-P cavity sensing part.

## Data Availability

Please contact the author’s team directly for data access.

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
