# Peer review of "Research on Transformer Omnidirectional Partial Discharge Ultrasound Sensing Method Combining F-P Cavity and FBG"

_sensors, 2023, doi:10.3390/s23249642_

Round 1

Reviewer 1 Report (Previous Reviewer 2)

Comments and Suggestions for Authors

Referee comments have been considered and the manuscript has been improved, considerably.

Author Response

Your previous comments have played an important role in refining this work. Thank you very much for your support.

Reviewer 2 Report (New Reviewer)

Comments and Suggestions for Authors

Nice work.  Would only recommend considering making some comments about ways to improve pD detection, and overall acoustic frequency response of the omnidirectional device.

Author Response

Thank you very much for your comments, we have included a paragraph at the end of this article: In the future, the introduction of more diaphragms with different resonant frequencies can be considered to improve the sensor's ability to cover the typical acoustic frequencies of PDs. In addition, by combining artificial intelligence algorithms and array signal processing algorithms, this sensor is expected to be used in localization studies of PDs.

This manuscript is a resubmission of an earlier submission. The following is a list of the peer review reports and author responses from that submission.

Round 1

Reviewer 1 Report

Comments and Suggestions for Authors

This paper study of Fiber Optic Omnidirectional Sensing for Partial Dis-2 charge in Transformers Combined with FBG and FP CavitySome points need to be addressed further:

1) Connection of FBG with FP is not so clear in this paper and need to be elaborated.

2) Specification of FBG like types of FBG, Apodization, chirp factor, index constrast etc need to highlighted?

3) For ultrasound detection process in Fig.1 FBG is etched or some other mechanism is adopted. justify.

4) FP characteristics like absorption of each mirrors, reflection and transmission of each mirror need to be mentioned.

5) Simualtion implementation like Fig. 3 need to be discussed?

6) Origin of formula (4) need to be addressed in detailed.

7) Fig. 11, the test results need to be addressed with effect of environment factors like vibration, acoustic noise etc.

Reviewer 2 Report

Comments and Suggestions for Authors
